# Oesophageal pH-Impedance for the Diagnosis of Gastro-Oesophageal Reflux Disease: Validation of General Population Reference Values in Children with Chronic Neurological Impairments

**DOI:** 10.3390/jcm10153351

**Published:** 2021-07-29

**Authors:** Anna Plocek, Beata Gębora-Kowalska, Wojciech Fendler, Ewa Toporowska-Kowalska

**Affiliations:** 1Department of Pediatric Allergology, Gastroenterology and Nutrition, Medical University of Lodz, 90-419 Lodz, Poland; beata.gebora-kowalska@umed.lodz.pl (B.G.-K.); ewa.toporowska-kowalska@umed.lodz.pl (E.T.-K.); 2Department of Radiation Oncology, Dana-Farber Cancer Institute, Harvard Medical School, Boston, MA 02115, USA; wojciech.fendler@umed.lodz.pl; 3Department of Biostatistics and Translational Medicine, Medical University of Lodz, 90-419 Lodz, Poland

**Keywords:** children, neurological impairment, gastro-oesophageal reflux disease, pH-impedance, oesophagogastroduodenoscopy

## Abstract

Diagnosis of gastro-oesophageal reflux disease (GORD) in children with chronic neurological impairment (NI) remains a clinical challenge. The study aimed to validate the relevance of the reference values used to assess gastro-oesophageal reflux (GOR) in children with NI and to determine the optimal cut-off level of the pH-impedance parameter with the best predictive value for outcomes associated with endoscopic assessments of the oesophagus. Sixty-seven children (32 male, 35 female; age: interquartile range, 5 years 6 months–14 years 10 months; median, 11 years 3 months) with NI were prospectively recruited for the study. The exclusion criteria were previous fundoplication and lack of consent for the study. All patients underwent evaluations for GOR disease, including pH-impedance and gastroscopy. Based on endoscopy, oesophagitis was diagnosed in 22/67 children (32.8%); 9/67 (13.4%) were classified as having Hetzel–Dent grade III or IV. GOR was present in 18/67 children (26.9%), as determined by pH-impedance. Patients with endoscopic lesions had a significantly higher number of total reflux (*p* = 0.0404) and acidic episodes (*p* = 0.0219). The total number of reflux episodes, with a cut-off level of 44 episodes, was the pH-impedance parameter most strongly predictive of the presence of lesions in gastroscopy (specificity: 50%, sensitivity: 73%). These findings suggest that endoscopic lesions may be present in children with chronic NI with a low number of GOR episodes, as recorded by pH-impedance. The use of standardised reference norms determined for the general population may underestimate GOR episodes in this group of patients.

## 1. Introduction

Multichannel intraluminal impedance-pH (MII-pH), also called pH-impedance, is a valuable diagnostic tool in the evaluation of gastro-oesophageal reflux (GOR) disease (GORD) and is particularly recommended for assessing the correlation between clinical manifestations and the physiochemical properties of reflux, in addition to investigating the role of various types of refluxes in the aetiology and pathogenesis of oesophagitis and other signs and symptoms of reflux disease [1]. The method was developed in the 1990s at Helmholtz Institute in Aachen, Germany, and was first tested in clinical trials in 2002 [2]. The test’s concept is based on the intraoesophageal measurement of impedance changes between neighbouring pairs of sensors integrated within a single probe using a pH sensor. Measurement of intraoesophageal pH-impedance provides information on the physiochemical nature of GOR irrespective of pH, thereby increasing the diagnostic sensitivity of the study compared to that of conventional pH-metry [3]. A protocol standardising the principles and interpretation of MII-pH tracing in children was presented in the position statement by the European Paediatric Impedance Working Group (EURO-PIG) [4]. The document quotes reference values determined based on studies carried out in healthy adults [2,5,6]. The position statement on the indications, methods, and interpretation of pH-impedance in children, which adopted recommendations of EURO-PIG, was published by the British Society of Paediatric Gastroenterology, Hepatology and Nutrition in 2017 [7]. Although pH-impedance offers the possibility to precisely characterise and quantitatively assess GOR, the correlation between testing results and the endoscopic assessment of the oesophagus in children remains ambiguous and is based on several studies investigating clinically diverse groups of patients, with or without a neurological impairment (NI) [8,9,10]. It was shown that, in children with erosive oesophagitis, the number of specific reflux types recorded using the pH-impedance method was higher than in those with non-erosive reflux disease, while the qualitative interpretation of MII-pH did not differ between the compared groups of patients [11]. These discrepancies may have resulted from having used imperfect norms.

Our study aimed to validate the relevance of the reference values currently used to assess GOR in a group of children with chronic NI. We also attempted to determine cut-off points in the quantitative analysis of pH-impedance tracing for the best predictive value for the outcomes of endoscopic assessments of the oesophagus in the study population.

## 2. Materials and Methods

### 2.1. Participants

This prospective study included 67 children with consecutive chronic NI (32 male, 35 female; age: interquartile range (IQR), 5 years 6 months–14 years 10 months; median, 11 years 3 months) who were hospitalised in the Department of Paediatric Gastroenterology at the Medical University of Lodz between 2015 and 2018. Children were eligible for this study if they met the following criteria: (1) medical diagnosis of permanent NI, (2) aged 4–18 years old, (3) Gross Motor Function Classification System score IV or V [12], and (4) completed assessment. Forty-five children were on scheduled anticonvulsant therapy for epilepsy. Exclusion criteria were previous fundoplication and lack of consent for participation in this study. Diagnoses and clinical characteristics of the subjects at the time of entry into the study are shown in Table 1.

All patients underwent an evaluation for GORD, including pH-impedance and upper gastrointestinal (GI) endoscopy. 

The study protocol was approved by the Bioethics Committee of the Medical University in Lodz (No. RNN/152/08/KE). Written informed consent was obtained from the children’s parents before the study.

### 2.2. Multichannel Intraluminal Impedance Protocol

All patients underwent a 24-h MII-pH (SLEUTH Sandhill Scientific, Denver, CO, USA). In all participants, the same protocol for the 24-h MII-pH measurements was used: the individuals fasted for 4–6 h before the test; the MII-pH was recorded by a portable data logger and a combined pH-impedance catheter. Three types of catheters were deployed: (1) infant (for patients with a height <75 cm) and (2) paediatric (for patients with a height <150 cm) catheters with a diameter of 2.13 mm/6.4 Fr with 6 impedance channels and 1 pH sensor positioned in the centre of the most distal impedance channel (for infants and children with a height of <150 cm); and (3) catheters for adults (for children with a height of >150 cm) with a diameter of 2.3 mm/6.9 Fr with 6 impedance channels and 1 pH sensor positioned in the centre of the second most distal impedance channel. In infant catheters, impedance rings were 1.5 cm apart; in paediatric and adult catheters, the rings were 2 cm apart.

The catheters were pre-calibrated in buffers with a pH of 4 and 7 and inserted via the anterior nares. According to the ESPGHAN EURO-PIG standard protocol, the pH-measuring sensor was located 2 vertebrae above the diaphragm [4]. We used Strobel’s formula to estimate the position of the pH probe [13]. A chest X-ray was performed to ensure correct pH sensor placement. The 6 impedance and pH signals were recorded at 50 Hz (every 0.02 s). The traces were analysed using BioView Analysis (Sandhill Scientific Inc. Denver, CO, USA), and all recordings were manually verified.

The patients did not receive prokinetic drugs or drugs that decreased gastric secretion. Proton pump inhibitors, histamine H2-receptor blockers, and prokinetic drugs were stopped 7, 3, and 2 days prior to the test, respectively [14]. An anticonvulsant treatment regimen was sustained during impedance monitoring.

Interpretation of the MII-pH data was based on the *Porto Consensus on Detection and Definitions of GOR* [15], with modifications published by Zerbib et al. [5].

It was determined that a liquid reflux episode could be diagnosed based on MII-pH once a retrograde decrease in intraoesophageal impedance of ≥50% from the baseline in ≥2 distal impedance channels occurred (impedance-detectable event). Only liquid refluxes lasting at least 3 s were taken into account. An acid reflux episode was defined as a decrease in the pH level ranging from an initial value of >4.0 to a value of <4.0 during the physical presence of refluxate (as confirmed by impedance sensors). GOR incidents for which the pH value ranged from 4 to 7 were classified as weakly acidic refluxes, while those with a pH ≥ 7 were classified as weakly alkaline. Acid reflux index (total oesophageal acid exposure) was defined as the percentage of recording time with pH < 4 (the normal values are up to 7% for children over 1 year of age, as accepted by the North American Society of Paediatric Gastroenterology, Hepatology and Nutrition and the European Society of Pediatric Gastroenterology, Hepatology and Nutrition) [16]. An MII-pH result was considered positive if either the number of reflux events per 24-h monitoring period was ≥75, the number of acidic events was ≥50, the number of weakly acidic reflux events was ≥33, the number of weakly alkaline events was ≥15, or in combination [5].

### 2.3. Oesophagogastroduodenoscopy (OGD)

All individuals underwent OGD combined with an oesophageal mucosa biopsy. Biopsies were obtained at approximately 3 cm above cardia and from macroscopically diseased areas. The OGD procedure was usually carried out on the day following the MII-pH assessment; in two subjects, OGD was performed within 2 days from reflux monitoring. In every case, the endoscopic examinations were performed under general anaesthesia.

The appearance of oesophageal mucosa was graded based on the Hetzel–Dent scale [17]. Histological criteria of oesophagitis include basal zone hyperplasia, papillary lengthening, or either an increased number of neutrophils, lymphocytes, or both.

### 2.4. Statistical Analysis

In the first stage, we constructed dot-plots for the analysed reflux episode numbers and compared their distribution with the percentile ranges defined by Zerbib et al. [5]. The comparison was aimed at detecting an over/underrepresentation of patients with and without reflux episodes within the respective ranges of 0–25%, 25–75%, 75–95%, and >95%. A Fisher’s exact test with a Freeman–Halton correction was used for this purpose.

Second, we compared the numbers of different types of reflux episodes between children with and without oesophageal lesions (Hetzel–Dent II–IV and 0–I, respectively) detectable on gastroscopy and compared that with the expected numbers estimated based on Zerbib data.

Finally, for variables that differed significantly between the groups, receiver operating characteristic (ROC) curves were constructed to determine the diagnostic potential and optimal cut-off values for detection/anticipation of oesophagitis (lesions confirmed by endoscopy). Boundaries of 95% confidence intervals (CI) were computed for areas under the ROC curves (AUC), while *p* levels were computed for pairwise comparisons, with *p* < 0.05 considered statistically significant.

Based on the Pediatric Z-Score Calculator provided by the Children’s Hospital of Philadelphia, a Z-score was calculated for all patients [18].

## 3. Results

Based on OGD, oesophagitis of any severity was diagnosed in 22/67 children (32.8%), and 9/67 (13.4%) were classified as Hetzel–Dent grade III or IV. One patient (1.5%) was diagnosed with oesophageal stenosis and another (1.5%) with eosinophilic oesophagitis.

The mean duration of the MII-pH was 23.47 ± 2.62 h. Based on the pH sensor, a positive acid reflux index (RI > 7%) was found in 19/67 children (28.4%). GOR, as determined by MII-pH, was present in 18/67 (26.9%) participants. Acid, non-acid, weakly acidic, and weakly alkaline refluxes were present in 7/67 (10.4%), 13/67 (19.4%), 13/67 (19.4%), and 2/67 (3.0%) children, respectively, with the presence of more than one type of GOR in 14/67 (20.9%) patients. Individual variability in MII-pH parameters is depicted in Table 2.

### 3.1. Comparison of Reflux Numbers

We compared the number of different types of reflux episodes registered by MII-pH in our group of patients with NI, with expected frequencies based on the reference ranges defined by Zerbib (Table 3). The distribution observed in our patients was significantly different from that in the reference study, with more patients showing extremely high (>95%) numbers of total, acidic, and weakly acidic reflux episodes. In all cases, belonging to a particular predefined range from the reference cohort was not associated in any way with the presence of lesions detectable on gastroscopy (Table 4). Patients with endoscopic lesions had higher numbers of total reflux episodes and higher counts of acidic episodes than those without lesions (*p* = 0.0404 and *p* = 0.0219) (Figure 1). The number of weakly acidic reflux episodes was also higher in patients with lesions detectable in OGD, but the difference was not statistically significant (*p* = 0.1452). Additionally, the bolus exposure index was higher in patients with oesophagitis (*p* = 0.043). Patients with and without lesions, as assessed by OGD, did not differ significantly in terms of sex distribution (11 girls and 11 boys vs. 21 girls and 24 boys, *p* = 0.9969) or age (114 (25–75%, 61–177) vs. 139 (25–75%, 79–180) months, *p* = 0.3604).

### 3.2. Evaluation of the Diagnostic Potential of pH-Impedance

Determination of the optimal cut-off levels of pH-impedance parameters for predicting the presence of lesions detectable on gastroscopy proved difficult, with the AUCs being above the threshold of significance but performing poorly in terms of clinical applicability. Nevertheless, the total number of reflux episodes was associated with an ROC curve with an AUC of 0.66 (95% CI, 0.52–0.80), with a cut-off level of 44 episodes providing the best diagnostic threshold with a specificity of 50% and a sensitivity of 73%. The number of acidic episodes performed similarly with an AUC of 0.67 (95% CI, 0.53–0.81). In the latter case, the cut-off with the highest diagnostic accuracy was 28 episodes of acidic reflux (sensitivity: 46%, specificity: 78%). The respective ROC curves are presented in Figure 2 and Figure 3.

## 4. Discussion

Despite the availability of various diagnostic methods, the diagnosis of GORD in children with chronic NI remains a considerable clinical challenge. Given the deficits in verbal communication, a diagnosis based on the typical clinical manifestations is rarely possible. Objective confirmation of oesophagitis is provided by upper GI endoscopy with a biopsy, which, in addition to erosive lesions, also allows for the detection of oesophageal strictures, Barrett’s oesophagus, and eosinophilic oesophagitis. It is, however, an invasive test, and, in children with NI, it usually requires general anaesthesia, whose risk, given the accompanying respiratory co-morbidities, is higher than that in the general population [19]. Both the European Society of Paediatric Gastroenterology, Hepatology and Nutrition and the North American Society of Pediatric Gastroenterology, Hepatology, and Nutrition guidelines consider it acceptable to empirically use an antisecretory treatment, but emphasise the need for regular monitoring of its efficacy [1]. Performance of diagnostic studies objectively assessing GORD (pH-metry alone or either MII-pH, upper GI endoscopy, or in combination) is recommended both at the point of making treatment decisions and over the course of the long-term monitoring of treatment effects.

Studies published so far have demonstrated a low predictive value of MII-pH in predicting oesophageal changes in children, while data on correlation with clinical manifestations are ambiguous. In the study by Dalby et al., the total number of reflux episodes detected by MII-pH in children with GERD and endoscopically confirmed oesophagitis remained within normal values [20]. Hojsak et al., analysed pH-impedance tracings and upper GI endoscopy results of 77 children without NI and determined that the numbers of various GOR episodes (total, non-acidic, and poorly alkaline) were significantly higher in children with oesophagitis; however, the rate of abnormal MII-pH tracing according to the adopted criteria was comparable between patients with and without endoscopic changes (34.7% vs. 22.7%, *p* = 0.07) [11]. The authors did not observe any differences in the indices that correlated the parameters of pH-impedance tracings with clinical manifestations (symptom index, symptom sensitivity index, symptom association probability). In contrast, Rossi et al. reported the clinical appropriability of the MII-pH procedure in the diagnosis of GORD in children, confirmed by high rate of clinical response to the PPI therapy [21]. The data focusing on the characteristics of GORD based on MII-pH in children with NI are limited, furthermore the available studies differ with respect to the methodology and are based on small numbers of patients [10,22,23,24]. Del Buono et al., analysed the MII -pH tracings with respect to the mode of feeding—by mouth or naso-gastrically [22]. The authors reported a trend towards more GOR events (*p* = 0.628) in individuals on tube feeding. This observation was not reproduced by the study of our group investigating the influence of percutaneous endoscopic gastrostomy on GOR exponents in similar group of patients—316 reflux episodes were recorded in 10 orally fed children compared with 90 in five individuals on tube feeding [23]. Notably, the results of study indicated the higher proportion of acid refluxes in children with NI and suggested that even if the pH-MII numerical results remain within the “normal” range, GOR still may be potentially harmful to the oesophageal mucosa.

The attempt to characterize the properties of GOR in paediatric patients with NI was also undertaken by Kawahara et al. [24]. The authors discriminated the patients into two groups (“normal” and “abnormal” GOR) based on pH reflux index and total number of impedance reflux episodes and compared the distribution of particular reflux types between the subgroups. In children with “abnormal” GOR the number of all types and acid refluxes, recorded both distally and proximally, was significantly higher as compared to subjects with MII-pH interpreted as “normal”. No correlation with gastroscopy nor clinical exponents of GOR were included which limited the comparison of data provided by this study with our findings. Fukahori et al., (in a study primarily assessing the role of baseline impedance) retrospectively assessed pH-impedance tracings in 14 children with chronic NI (7 patients with erosive oesophagitis and 7 without endoscopic changes) and found no differences between the groups for any of the reflux types [10]. However, the authors did not provide the percentage of pathological tracings in the subgroups, and the group in which endoscopic results were available was limited in size. The same authors investigated the relationship between MII-pH parameters and pH index in a group of 61 children with NI (35 underwent gastroscopy and 9 were diagnosed with oesophagitis). They reported a significantly higher percentage of esophagitis among patients with >70 reflux episodes than those with ≤70 episodes [25].

The norms provided by Zerbib et al. [5], Shay et al. [2], and the EURO-PIG [4] are used in the interpretation of MII-pH in children. For ethical reasons, none of the adopted paediatric reference values were determined by examining a homogenous population of healthy children, which may have explained the poorer correlation of the MII-pH tracing results with the clinical and endoscopic manifestations of GORD in children, compared to the respective results for adults [9,26]. Cresi et al. overcame this problem and considered symptomatic infants and children with negative MII-pH results. They described the distribution of MII-pH values in children suspected of GORD with normally acidic GOR exposure and no association between GOR events and symptoms [27]. The German Paediatric Impedance Group and the Japanese Pediatric Impedance Working Group recommend the adoption of >70 GOR episodes per 24 h in children aged >1 year and >100 in children aged <1 year as abnormal values that suggest a positive pH-impedance [28,29] (versus >75 given by Zerbib et al. [5]). The cut-off points were determined based on a retrospective review of data obtained from a total of 700 patients examined at various centres and representing various clinical conditions and indications for work-up [28]. For the sake of comparison, the Lyon Consensus (2017) adopts <40 episodes detected by impedance to be normal (with and without proton pump inhibitor treatment), and >80 episodes to be pathological, and classifying the intermediate values as inconclusive [30]. The pH-impedance tracings that we analysed in conjunction with the endoscopic assessment of the oesophagus were obtained from a group of 67 children with chronic NI prospectively studied according to a unified study protocol [5,15]. The study population was uniform in terms of severity of gross motor impairment (classified at level IV and V of the Gross Motor Function Classification System), nutritional status, and the nutritional model and was therefore representative of the population. Moreover, the visual assessment of pH-impedance tracings, which preceded automated analysis, was carried out by the same investigators, who had ample experience in performing this procedure. 

To validate the reference values (Zerbib et al. [5]), we used point diagrams, which allowed us to assess the distribution of specific types of refluxes recorded by pH-impedance in our cohort at percentile intervals defined by the authors cited above. We showed that the distribution of pH-impedance tracing parameters in children with chronic NI significantly differed from that in healthy adults. At the same time, being in a particular predefined range from the reference cohort did not correlate with the results of the endoscopic oesophageal assessment. These results suggest that the norms proposed by Zerbib et al., may not be appropriate for the interpretation of pH-impedance in children with NI. 

We also observed that children with oesophagitis are characterised by significantly higher numbers of total and weakly acidic GOR episodes than children with normal endoscopies. In addition, the number of weakly alkaline GOR episodes was higher in the group of children with endoscopic changes, although these differences were not statistically significant. These findings are consistent with those of Hojsak et al., in children without NI and confirm the relationship between oesophagitis and the severity of GOR (total number of reflux episodes and number of acidic refluxes) [11]. The fact that the number of weakly acidic GOR episodes, but not the number of acidic episodes, better differentiated children with from those without erosive lesions in the oesophagus may be due to the diet consumed by the study subjects and the method of feeding. All children received commercially available liquid diets administered orally or using a feeding tube. This method of feeding reflects the real-life conditions of patients with severe NI and may induce higher pH of the reflux material and, consequently, the pH-impedance tracings. The relationship between gastric content pH and the type of food was well documented in a small randomised study of healthy volunteers [31], in which both the oral and the duodenal routes of administration of a liquid diet (polymeric or hydrolysed) were reported to result in higher intragastric pH compared to that obtained by a natural diet.

Based on our results, we propose adopting separate norms for interpreting pH-impedance tracing in children with NI, as calculated from the ROC curves in our study. Our analysis has demonstrated that the MII-pH parameter that best predicts the development of erosive lesions in the oesophagus (at a sensitivity of 75%) is the total number of GOR episodes detected by an impedance exceeding 44. This is a value very close to the one proposed in the Lyon Consensus as the limit of norm (40 episodes) [30]. A less sensitive but more specific parameter is the number of weakly acidic GOR episodes exceeding 28 (at a sensitivity of 46% and a specificity of 76%). Notably, the parameters for the quantitative assessment of pH-impedance in children with NI that we propose also largely depart from the paediatric norms provided by the German Paediatric Impedance Group [28]. The latter were based on the results of MII-pH tracings obtained, among others, from 46 “neurological” patients (with a chronic central nervous system condition as an inclusion criterion), in whom pathological tracings of pH, impedance, and pH-impedance were observed in 9%, 48%, and 43% of the subjects, respectively. The mean age of these patients was 0.5 years (range: 3 weeks–15 years). Therefore, these patients were less numerous and more diverse in terms of age compared with our cohort (67 patients; IQR 66–135 months). This study, however, did not provide absolute values for the numbers of specific GOR types, making any comparisons in this respect impossible. 

A different distribution of GOR types recorded by pH-impedance and demonstrated in this study in children with NI compared to the general population may explain the low predictive value of this method in predicting endoscopic oesophageal changes using the current quantitative norms, thereby defining a boundary that separates normal from pathological tracings [5]. In the analysed group, we found not only an over-representation of the subgroup with a high number of GOR episodes but also the presence of oesophageal lesions in children with fewer GOR episodes than normal, according to the norms proposed by Zerbib et al. [5]. In neurologically impaired patients, oropharyngeal dysphagia of central origin and GI dysmotility of complex aetiology form a “GOR-dysphagia complex” responsible for malnutrition and GORD and for damage to the respiratory tract caused by aspiration [32]. In addition, abnormalities in body stature, which are common in children with NI (multiplanar curvature of the spine, displacement of internal organs, spasticity, excessive flaccidity, lack of control over the head position, limb contractures), long-term immobilisation in the horizontal position (also during feeding), abnormal gastric emptying, GOR, and excessive loss of saliva, secondarily result in impairment of oesophageal clearance [33]. Thus, patients with NI are characterised by a reduced resistance of the oesophageal mucosal barrier, in whom inflammatory lesions can develop with less severity of reflux exposure. 

When assessing this study’s results, it is important to consider certain limitations. While searching for parameters that better correlate with endoscopic lesions of the oesophagus than the quantitative analysis of reflux types does, some researchers propose the so-called baseline impedance (BI). BI is a derivative of ion concentrations in the environment in which the measurement is performed. Experimental studies have revealed that BI decreases after acidic oesophageal perfusion [34]. This parameter is therefore a net product of the quantitative and qualitative severity of GOR and of the efficiency of oesophageal clearance. Based on a manual assessment of a typical pH-impedance tracing, Salvatore et al. demonstrated that BI is significantly lower in children with erosive oesophagitis [35], while Borrelli et al. failed to confirm this finding [36]. Assessment of the baseline impedance is recommended by some authors [7,29,30]; however, its automatic analysis is not possible with all the pH-impedance systems currently in use. Analysis of BI was not included in our study (the study design was based on the guidelines for the interpretation of pH-impedance in children and adults), which may constitute its limitation. The different ways of feeding patients in our study group (38 children were fed orally and 29 with nasogastric tubes) is another limitation. This could have influenced the parameters determined by MII-pH. Additionally, we interpreted GOR in NI children in accordance with reference values defined by Zerbib et al., which state the normal values for healthy adult subjects [5]. We were not able to refer the results of MII-pH tracings obtained in our children to paediatric norms because they do not exist. EURO-PIG provides only the cut-off value for the total number of refluxes [4], whereas Zerbib et al. provide us with quantitative data on all reflux categories [5]. Another possible limitation would be the lack of the evaluation of the interobserver reproducibility of the MII-pH interpretation. Intraprocedure and interobserver reproducibility of pH-IM interpretation was the aim of several studies. For example, Dalby et al. [37] investigated the reproducibility of reflux parameter obtained by 2 consecutive 24 h pH/MII monitoring periods in children and infants with symptoms of GERD and found that the reproducibility of the GERD diagnosis as based on the NASPGHAN criteria for RI was 77%.

## 5. Conclusions

In conclusion, our study results draw attention to a different distribution of the numbers and high variability in reflux episodes’ characteristics in children with chronic NI, compared to those in the general population. They demonstrate that the use of reference norms determined for the general population may underestimate GOR episodes in this group of patients. Our results also point to the need for a comprehensive work-up in patients with chronic NI with suspected GORD, as endoscopic lesions may also be present in individuals with a small number of GOR episodes recorded by pH-impedance.

## Figures and Tables

**Figure 1 jcm-10-03351-f001:**
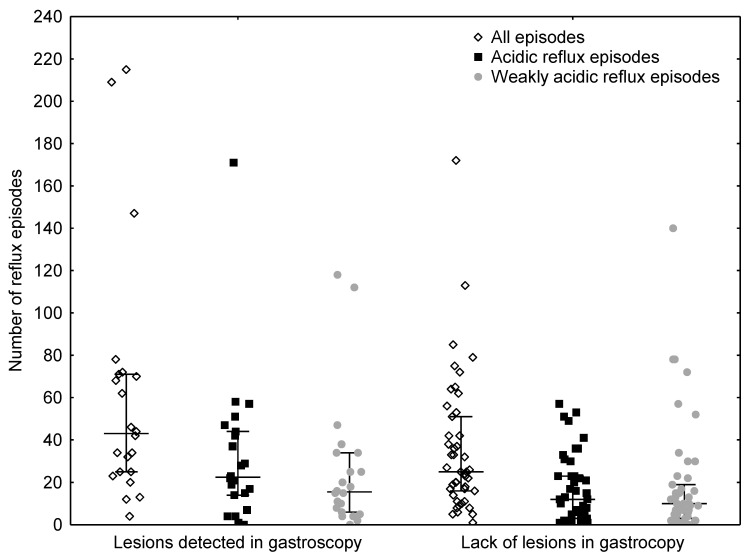
Number of reflux episodes in children with and without lesions in gastroscopy.

**Figure 2 jcm-10-03351-f002:**
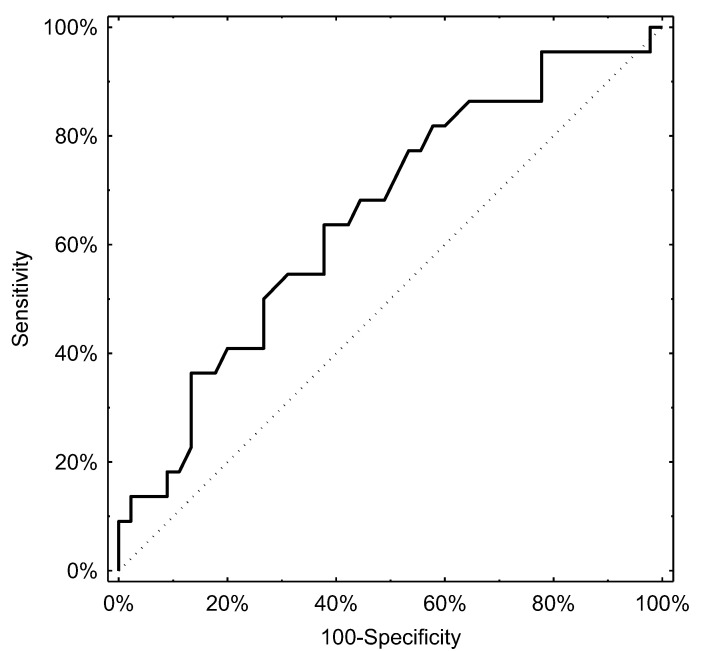
ROC curve derived from the total number of reflux episodes and lesions in gastroscopy.

**Figure 3 jcm-10-03351-f003:**
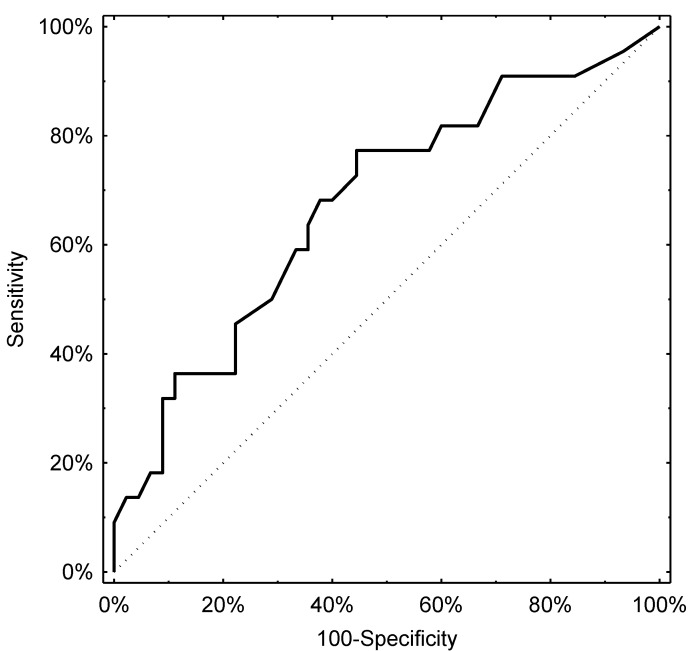
ROC curve derived from the number of acid reflux episodes and lesions in gastroscopy.

**Table 1 jcm-10-03351-t001:** Demographic characteristic of the study group.

	N of Patients	% of Patients
**Clinical Diagnosis**		
Cerebral palsy	37	55.2
Ceroid lipofuscinosis	5	7.5
Encephalopathy following brain tumour	2	3.0
Encephalopathy congenital	12	17.9
Encephalopathy acquired: post-traumatic, post-inflammatory	3	4.5
Aicardi syndrome	1	1.5
Neurodegenerative disorders due to congenital metabolic defects	7	10.4
**Feeding**		
Orally fed	38	56.7
Nasogastric tube	29	43.3
**Degree of Gross Motor Disability** (GMFCS)		
Level IV	60	89.6
Level V	7	10.4
**Nutritional Status**		
Severe undernutrition (BMI Z-score < −3)	37	55.2
Undernutrition (BMI Z-score > −3 < −1	22	32.8
Normal nutrition (BMI Z-score 0)	6	9.0
Overnutrition (BMI Z-score > 3)	2	3.0

GMFCS, Gross Motor Function Classification System; BMI, body mass index; N, number.

**Table 2 jcm-10-03351-t002:** GOR parameters determined by MII-pH.

Indices of MII-pH	Number (Median; Range)
reflux events	2981 (32; 1–215)
acidic reflux events	1437 (15; 0–171)
non-acid reflux events:	1544 (2; 0–140)
weakly acidic reflux events	1432 (11; 0–140)
weakly alkaline reflux events	112 (0; 0–45)
	**Index (Median; Range)**
Acid reflux index	5.75% (2.8%; 0–29.1%)
Bolus exposure index	2.74% (1.5%; 0.2–18.9%)
	**Time (Median; Range)**
Mean acid clearance time	334.36 s (126 s; 0–7177 s)
Mean bolus clearance time	14.39 s (12 s; 2–62 s)

GOR, gastro-oesophageal reflux; MII-pH, multichannel intraluminal impedance-pH.

**Table 3 jcm-10-03351-t003:** Comparison of the number of different reflux episode types registered by MII-pH in patients with NI with the frequencies expected on the basis of the reference ranges defined by Zerbib [4].

Range	Expected		Total		Acidic Reflux		Weakly Acidic Reflux		Alkaline Reflux	
	*N*	%	*N*	%	*N*	%	*N*	%	*N*	%
≤25%	16.75	25.00	26	38.81	26	38.81	17	25.37	46	68.66
25–50%	16.75	25.00	18	26.87	15	22.39	16	23.88	13	19.40
50–75%	16.75	25.00	5	7.46	11	16.42	12	17.91	5	7.46
75–95%	13.40	20.00	9	13.43	8	11.94	9	13.43	1	1.49
≥95%	3.35	5.00	9	13.43	7	10.45	13	19.40	2	2.99
*p* level	reference		0.0160		0.1135		0.0965		<0.0001	

NI, neurological impairment; MII-pH, multichannel intraluminal impedance-pH; *N*, number.

**Table 4 jcm-10-03351-t004:** Comparison of the numbers of different reflux episode types with (Hetzel–Dent II–IV) and without (Hetzel–Dent 0–I) detectable lesions in gastroscopy to the percentile ranges defined by Zerbib [4].

	Total		Acidic Reflux		Weakly Acidic Reflux		Alkaline Reflux	
Value range	No lesions	Lesions present	No lesions	Lesions present	No lesions	Lesions present	No lesions	Lesions present
≤25%	21 (46.67%)	5 (22.73%)	21 (46.67%)	5 (22.73%)	13 (28.89%)	4 (18.18%)	31 (68.89%)	15 (68.18%)
25–50%	12 (26.67%)	6 (27.27%)	10 (22.22%)	5 (22.73%)	12 (26.67%)	4 (18.18%)	10 (22.22%)	3 (13.64%)
50–75%	3 (6.67%)	2 (9.09%)	7 (15.56%)	4 (18.18%)	8 (17.78%)	4 (18.18%)	3 (6.67%)	2 (9.09%)
75–95%	4 (8.89%)	5 (22.73%)	4 (8.89%)	4 (18.18%)	5 (11.11%)	4 (18.18%)	0 (0.00%)	1 (4.55%)
≥95%	5 (11.11%)	4 (18.18%)	3 (6.67%)	4 (18.18%)	7 (15.56%)	6 (27.27%)	1 (2.22%)	1 (4.55%)
*p*-value	0.2917		0.2684		0.6075		0.5593	

## Data Availability

The data supporting reported results are available from the corresponding author on reasonable request.

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
