# Peer review of "Oesophageal pH-Impedance for the Diagnosis of Gastro-Oesophageal Reflux Disease: Validation of General Population Reference Values in Children with Chronic Neurological Impairments"

_jcm, 2021, doi:10.3390/jcm10153351_

Round 1

Reviewer 1 Report

I’ve read with attention the paper of Plocek et al. that is potentially of interest. The background and aim of the study have been clearly defined. The methodology applied is overall correct, the results are reliable and adequately discussed. I’ve only some minor comments:

  • It appears that no sample size calculation has been done. Should the authors confirm the the number of subjects enrolled is sufficient to support their conclusion?
  • The discussion is very long, repeating the most part of the results while not specifically focusing on the main ones, nor discussing the study limitation. It could be improved

Author Response

Dear Reviewer,

Thank you very much for the careful reviewiewing of our manuscript and valuable comments. 

Comment No 1: It appears that no sample size calculation has been done. Should the authors confirm that the number of subjects enrolled is sufficient to support their conclusion?

Answer: With regard to the question on sample size calculation we wolud like to explain, that formal sample size analysis was not performed as we recruited all available patients in the Department in an exploratory analysis aimed at delineation of proper reference ranges (as advised by Prof. Wojciech Fendler, who was responsible for the statistics)

Comment No 2: The discussion is very long, repeating the most part of the results while not specifically focusing on the main ones, nor discussing the study limitation. It could be improved

Answer: Our goal was to provide the comprehensive discussion, unfortunately it seems to be  a bit lengthly. We updated  the discussion by including the comments on  studies pertaining to pH-MII in children with NI. We also reanalyzed the limitations of our study (page 17).

Nevertheless, the part of the results being in some way duplicated in the discussion section is this: “To validate the reference values (Zerbib et al. [5]), we used point diagrams, which allowed us to assess the distribution of specific types of reflux recorded by pH-impedance in our cohort at percentile intervals defined by the authors cited above. We showed that the distribution of pH-impedance tracing parameters in children with chronic NI significantly differed from that in healthy adults. At the same time, being in a particular predefined range from the reference cohort did not correlate with the results of the endoscopic oesophageal assessment”. As this statement reflects the main objective of our study we would prefer not to remove it. To additionally justify our perception we would mention that the two other Reviewers did not suggest the shortening of the discussion.

Yours sincerely,

Authors

Reviewer 2 Report

Executive Summary

The manuscript titled “Oesophageal pH-impedance for the diagnosis of gastro-oesophageal reflux disease: Validation of general population reference values in children with chronic neurological impairments” investigated the potential application of Multichannel intraluminal impedance-pH (MII-pH) as a new diagnostic tool for gastro-oesophageal reflux (GOR) disease (GORD). Overall, the manuscript is well written with strong and scientific discussion. The authors may perform minor revisions to further improve the quality of the manuscript.

Major Comments

There is no major comment.

Minor Comments

  • In table 2, it will be better if the authors can change “Median (range)” into “N, Median (range)” to be consistent with the data. Then change the first column into descriptive items, such as “reflux events”, “non-acid events”, etc.
  • In table 3, the “expected” group’s percentage is more than 100%.

Author Response

Dear Reviewer,

Thank you for your valuable comments. We reanalyzed the tables (2. & 3.) and made corrections as recommended by the  Reviewer:

Comment No 1: In table 2, it will be better if the authors can change “Median (range)” into “N, Median (range)” to be consistent with the data. Then change the first column into descriptive items, such as “reflux events”, “non-acid events”, etc.

Answer: Based on  the suggestions in the review, we rearranged the table 2 to make its content more readable. The data in the first column were limited to the descriptive data and numerical data were transfered into the second column, along with appropriate headings adjustments. However the parameters describing pH-impedance are of various nature and units (numbers, percentages, duration of particular events) what makes it difficult for the unification.

Comment No 2: In table 3, the “expected” group’s percentage is more than 100%.

Answer: In table 3 we corrected the obvious mistake ( "expected" group's percentage is more than 100% and is 105%). The number= 13.4 corresponds to 20% (not 25%). 

Yours sincerely,

Anna Plocek

Reviewer 3 Report

Summary

The relevance of MII-pH parameters for the clinical diagnosis of GORD in children with neurological implications is studied. The authors position the findings of this study against diagnostic norms evaluated on a different subset of the general population (Zerbib et al.). Comparisons with studies that evaluated similar cohorts are deficient. This study adhered to ethical standards, and applications of statistical analysis are appropriate. Data presentation is legible, and the quality of writing is satisfactory.

Recommendation

The manuscript is recommended to be revised and resubmitted. Specifically, discussions of the publications listed below should be included.

  1. PMID: 16954955
  2. PMID: 29724467
  3. PMID: 20504244
  4. PMID: 21752017
  5. PMID: 28808763

Specific Comments

  1. In the study conducted by Zerbib et al., only liquid reflux events lasting ≥3 seconds were considered. Was a liquid reflux event duration criterion implemented in this study? Is this variable of any significance? See reference PMID: 11525610.
  1. Evaluation of reproducibility of the impedance-pH recordings would have been a nice inclusion.

Author Response

Dear Reviewer,

Thank you very much for the careful reviewiewing of our manuscript and valuable comments. We analyzed them very carefully and the recommendations are included in the revised version of the manuscript:

Comment No 1: The manuscript is recommended to be revised and resubmitted. Specifically, discussions of the publications listed below should be included.

  1. PMID: 16954955
  2. PMID: 29724467
  3. PMID: 20504244
  4. PMID: 21752017
  5. PMID: 28808763

Answer:  We updated the references accordingly to the recommendations. We also included appropriate comments on their content in the discussion section (References: 20-24).

Comment No 2: In the study conducted by Zerbib et al., only liquid reflux events lasting ≥3 seconds were considered. Was a liquid reflux event duration criterion implemented in this study? Is this variable of any significance? See reference PMID: 11525610.

Answer: In our study the interpretation of the MII-pH data was based on the Porto Consensus on Detection and Definitions of GOR, with modifications published by Zerbib et al. According to the cited protocol only liquid reflux events lasting at least 3 seconds were taken into account. This criterion was added in the reviesed manuscript (section 2.2. Multichannel intraluminal impedance protocol).

Comment No 3: Evaluation of reproducibility of the impedance-pH recordings would have been a nice inclusion.

Answer: The  interobserver reproducibility analysis was not performed in our study protocol. Each tracing was analyzed simultaneously by skilled observers (BGK, AP, ETK) with nationally recognized experience in the field of pH-IM. We absolutely agree that the evaluation of the interobserver reproducibility of the procedure interpretation would be valuable, therefore we added appropriate comment in the revision,  denoting the lack of validation as a limitation of the study (page 17).

Yours sincerely,

Authors